# The Classification, Molecular Structure and Biological Biosynthesis of Flavonoids, and Their Roles in Biotic and Abiotic Stresses

**DOI:** 10.3390/molecules28083599

**Published:** 2023-04-20

**Authors:** Wei-Bing Zhuang, Yu-Hang Li, Xiao-Chun Shu, Yu-Ting Pu, Xiao-Jing Wang, Tao Wang, Zhong Wang

**Affiliations:** 1Jiangsu Key Laboratory for the Research and Utilization of Plant Resources, Institute of Botany, Jiangsu Province and Chinese Academy of Sciences (Nanjing Botanical Garden Mem. Sun Yat-Sen), Nanjing 210014, China; weibingzhuangnj@sina.com (W.-B.Z.); liyuh1998@163.com (Y.-H.L.); islbe@163.com (X.-C.S.); johnwt1007@163.com (T.W.); 2College of Tea Science, Guizhou University, Guiyang 550025, China; pyt177693@163.com

**Keywords:** abiotic stress, biotic stress, flavonoids, molecular mechanism, transgenic plants

## Abstract

With the climate constantly changing, plants suffer more frequently from various abiotic and biotic stresses. However, they have evolved biosynthetic machinery to survive in stressful environmental conditions. Flavonoids are involved in a variety of biological activities in plants, which can protect plants from different biotic (plant-parasitic nematodes, fungi and bacteria) and abiotic stresses (salt stress, drought stress, UV, higher and lower temperatures). Flavonoids contain several subgroups, including anthocyanidins, flavonols, flavones, flavanols, flavanones, chalcones, dihydrochalcones and dihydroflavonols, which are widely distributed in various plants. As the pathway of flavonoid biosynthesis has been well studied, many researchers have applied transgenic technologies in order to explore the molecular mechanism of genes associated with flavonoid biosynthesis; as such, many transgenic plants have shown a higher stress tolerance through the regulation of flavonoid content. In the present review, the classification, molecular structure and biological biosynthesis of flavonoids were summarized, and the roles of flavonoids under various forms of biotic and abiotic stress in plants were also included. In addition, the effect of applying genes associated with flavonoid biosynthesis on the enhancement of plant tolerance under various biotic and abiotic stresses was also discussed.

## 1. Introduction

Flavonoid compounds, one of the main classes of plant secondary metabolites, are commonly present in plants [1,2]. Over 9000 flavonoid derivatives have been identified in various plants, which are further divided into different subfamilies depending on the modification of their basic structure [3,4]. Some plant flavonoids may be used as dietary compounds, which can improve human health and prevent many diseases [5,6]. Based on the epidemiological, clinical and animal studies, flavonoids may exert beneficial effects on human health when an individual is suffering from various diseases, such as cardiovascular disease and cancer [7,8,9].

Phenolic compounds are a group of metabolites derived from the secondary pathways of plants that contains flavonoids, phenolic acids, tannins, lignans, and coumarins [10,11]. Among them, flavonoid compounds are naturally distributed in fruits, seeds, flowers and the leaves of plants, and play important roles in regulating the growth and development of plants [12,13]. Plant flavonoids are involved in pollination, auxin transport inhibition, male fertility, allelopathy, seed development and flower coloring [14,15,16]. Moreover, flavonoids have also been reported to play protective roles against abiotic stresses through ROS detoxification, including UV-B radiation, low temperatures, salt, drought, and heavy metal stress [12,17,18,19,20]. In addition, flavonoids also offer protection against biotic stresses, and are capable of participating in plant–microbe interaction; they are also especially involved in the symbioses of plants with rhizobia and mycorrhizal fungi, as well as in the interactions between pathogenic microorganisms and plants [21,22]. Some flavonoids (particularly anthocyanins) act as major flower pigments, which help plants attract pollinators [13].

Genetically modified (GM) plants are those whose genomes have been modified via the introduction of foreign DNA derived from bacteria, fungi, viruses, or animals; this process has been used to verify the gene function and generate GM plants [23,24]. Up to now, two common systems have been used for gene transfer in the generation of GM plants: the *Agrobacterium*-mediated system and the gene-gun-mediated system [25,26]. With the development of transgenic technology, genetic modification has been widely used to verify the gene functions in the flavonoid biosynthesis pathway, and the roles of flavonoids in plant development and response to (a)biotic stresses [27,28,29].

Flavonoids have attracted significant attention in recent years. This review summarizes the recent progress made in the research of flavonoids, including their classification, biosynthesis and mechanisms of responding to adversity. Moreover, the application of flavonoids in modern biotechnologies and the prospects of flavonoids in response to various stresses are also reviewed here.

## 2. Flavonoids Classification

Flavonoids, a diverse group of natural substances with variable phenolic structures, are widely distributed in plants [30]. So far, over 9000 flavonoids have been identified, which are further classified into seven subgroups depending on the modification of their basic structure; these subgroups include flavones, flavanones, isoflavones, flavonols or catechins, and anthocyanins [31,32,33,34].

### 2.1. Flavones

Flavones, one of the important subgroups of flavonoids, contain the backbone of 2-phenylchromen-4-one (2-phenyl-1-benzopyran-4-one) (Figure 1A), and are widely distributed in leaves, flowers, and fruits of many plants, such as celery, parsley, red pepper, chamomile, mint, and ginkgo [35,36,37,38,39,40]. Flavones possess the C2′-C3′ double bond and C-ring, bearing a ketone at position C4′ [41]. Most flavones have an A-ring, bearing a hydroxyl group at position C5’, with hydroxylation usually occurring at position C7′ of the A-ring and positions C3′ and C4′ of the B-ring [42,43]. In addition, the glycosylation of flavones mainly occurs at positions C5′ and C7′ of the B-ring, and their methylation and acylation happens in the B-ring of the hydroxyl groups [42]. The major flavones are apigenin, luteolin, baicalein, chrysin and their derivatives [44].

### 2.2. Flavonols

Flavonols, known as 3-hydroxyflavone, process a hydroxyl group at position C3′ of the C-ring (Figure 1B) and are glycosylated to some extent [45]. Flavonols are the building blocks of proanthocyanidins, which are abundant in vegetables and fruits such as onions, lettuce, tomatoes, apples, grapes and so on [46,47,48,49,50,51,52,53]. Glycosylation and methylation mainly contribute to the diversity of flavonol compounds [54]. O-glycoside, as the main substituent, occurs at position 7 of the A-ring and position 3 of the C-ring, and generates flavonol glycosides [55]; these mainly include kaempferol, quercetin, limocitrin, and isorhamnetin [56]. Over the past 15 years, the number of flavanones identified has significantly increased [57]. Dietary flavonols play important roles in human health due to their anti-oxidant, cardio-protective, anti-bacterial, anti-viral and anti-cancer activities [58].

### 2.3. Flavanones

Flavanones, also called dihydroflavones, have the saturated C-ring [59]. The unsaturated double bond between positions C2′ and C3′ of the C-ring is the only chemical structural difference between flavanones and other flavonoids (Figure 1C) [60]. Flavanones are mainly distributed in all citrus fruits [61,62], and have hydroxyl groups at positions C5′ and C7′ of the A-ring and have hydroxyl/methoxy substituents at positions C3′ or C4′ of the B-ring [63]. O-glycoside, as the main substituent, occurs at position 7 of the flavanone aglycone, and produces the flavanone glycosides [64]. The substituents of flavanone glycosides contain glucoside, rhamnoside, rutinoside, and neohesperidoside [65]. The flavanones can be divided into hesperitin, naringin, naringenin, eriodictyol, hesperidin, pinocembrin, and likvirtin based on their structural differences [66,67]. Among them, naringenin and hesperetin, as the main flavanones, are mainly present in lemons, oranges, limes, tangerines and grapefruit [61,68,69].

### 2.4. Isoflavonoids

Isoflavones contain a C-ring with a B-ring at position 3 (Figure 1D), which is the only chemical structural difference compared to other flavonoids [70]. Isoflavonoids are mainly distributed in leguminous plants [71], and play important roles in microbial signaling and nodule induction in legumes [72]. Isoflavones can be divided into two groups, genistein and daidzin, which exist in the chemical structures of aglycone, 7-*O*-glucoside, 6′-*O*-acetylglucoside, and 6′-*O*-malondialdehyde glucoside [73,74]. Isoflavones have strong antioxidant properties, which can reduce the damage caused by free radicals to plant cells and improve their resistance to UV-B radiation, salt stress and osmotic stress [75,76,77].

### 2.5. Neoflavonoids

There is a 4-phenyl coumarin backbone and no hydroxyl substitution at position C2′ in neoflavonoids (Figure 1E), which are rarely found in food plants [78]. Neoflavonoids are mainly divided into four substructure types based on their basic skeleton structure, namely dalbergia phenols, dalbergia quinones, dalbergia lactones, and benzoyl benzenes [79,80].

### 2.6. Flavanols, Flavan-3-ols or Catechins

Flavanols, known as dihydroflavonols/catechins, are the 3-hydroxy derivatives of flavanones (Figure 1F) [81] and are naturally occurring plant-based nutrients due to their antioxidant properties. Flavanols are also referred to flavan-3-ols, as the hydroxyl group is always bound to position 3 of the C-ring [82]. Unlike other flavonoids, flavanols have no double bond between positions C2′ and C3′ of the C-ring [83,84,85]. Flavanols are abundant in manly fruits, such as bananas, apples, blueberries, and pears [86,87,88,89], and are divided into several types: catechin, gallocatechin, catechin 3-gallat, gallocatechin 3-gallate, epicatechin, epicatechin 3-gallate, and epicatechin 3-gallate [90]. Flavanols and their metabolites play important roles in plant responses to various stresses due to their potent antioxidant and free radical scavenging activities [91,92].

### 2.7. Anthocyanins

Anthocyanins, the glycosylated polyphenolic compounds, are a group of soluble vacuolar pigments that generate a series of orange, red, purple and blue colors in vegetative and re-productive plant organs [93,94]. So far, over 650 anthocyanins have been identified in various plants [95] and grouped into the following six categories: cyanidin, delphinidin, malvidin, pelargonidin, peonidin, petunidin, and the corresponding derivatives [96,97]. Unlike other flavonoids, except flavanols, anthocyanins carry no ketone group at position 4 of the C-ring (Figure 1G). Anthocyanins are mainly found in the outer cell layer of various fruits and vegetables, such as black currants, grapes, and berries [98,99,100,101,102].

## 3. Flavonoid Biosynthesis in Plants

### 3.1. Regulation of Flavonoid Biosynthesis

The biosynthesis of flavonoids occurs at the convergence of the shikimate and acetate pathways, with the former producing p-coumaroyl-CoA and the latter involved in a C_2_ elongation reaction [103] (Figure 2). Phenylalanine ammonia lyase (PAL) catalyzes the deamination of phenylalanine into trans-cinnamic acid and ammonia; this is the first step of the phenylpropanoid pathway [104]. Next, cinnamic acid hydroxylase (C4H), a cytochrome P450-dependent hydroxylase, hydroxylates the trans-cinnamic acid to produce p-coumaric acid [105]. The 4-coumaric acid CoA ligase (4CL) catalyzes 4-coumaric acid to generate 4-coumaroyl CoA, which is a key intermediate in the biosynthesis of lignin and flavonoids [106]. Chalcone synthase (CHS) catalyzes the formation of naringenin chalcone from one p-coumaroyl CoA and three malonyl-CoA molecules [107]. Malonyl-CoA is the essential building block of natural products such as fatty acids, polyketides, and flavonoids [108]. Naringenin chalcone is converted to produce flavanones through chalcone isomerase (CHI), which is cyclized to produce naringenin under the catalytic action of CHI [109]. Naringenin, a general precursor of flavonoid compounds, is catalyzed by flavone synthase I and II (FNSI and II) and isoflavone synthase (IFS) to generate flavones and iso-flavones, respectively [110,111]. In addition, naringenin is also catalyzed by flavanone-3-hydroxylase (F3H), flavonol 3′-hydroxylase (F3′H), and flavonol 3′5′-hy-droxylase (F3′5′H) to produce dihy-drokaempferol, dihydroquercetin, and dihydromyricetin, respectively [112,113]. The dihydroflavonols are further converted to flavonols (kaempferol, quercetin, and myricetin) by flavonol synthase (FLS), which is also converted to leucoanthocyanidins under the catalysis of the dihydroflavonol 4-reductase (DFR) [114]. Anthocyanidins are then formed from leucoanthocyanidins by leucoanthocyanidin dioxygenase (LDOX), and are further catalyzed by uridine diphosphate (UDP)-glucose flavonoid-3-*O*-glycosyltransferase (UFGT) [115]. Leucoanthocyanidins and anthocyanidins can also be converted to proanthocyanidins by leucoanthocyanidin reductase (LAR) and anthocyanidin reductase (ANR), respectively [116,117,118]. Finally, anthocyanins are stabilized via the modification of glycosylation, methylation, and acylation [119].

### 3.2. Transcription Factors (TFs) Regulate Flavonoid Biosynthesis

Genes involved in flavonoid biosynthesis are classically divided into early biosynthetic genes (EBGs; *CHI*, *CHS*, *F3′H* and *F3H*) and late biosynthetic genes (LBGs; *FLS*, *DFR*, and *ANS*) [120,121,122]. Flavonoid biosynthesis is not only affected by plant hormones, but also regulated by various stresses [123]. These environmental or developmental regulations mostly rely on the regulatory effect of TFs on structural genes in the flavonoid biosynthesis pathway (EBGs and LBGs) [124]. Based on the differences in the DNA-binding characteristics, several families of TFs have been described as regulators of flavonoid biosynthesis and metabolism in many plants; these include MYB, bHLH, WD40, bZIP, NAC, MADS box, Dof, and WRKY [125,126]. MYB proteins include highly conserved N-terminal MYB repeats (1R, R2R3, 3R, and atypical) [127,128]. The flavonoid pathway genes are predominantly regulated by the R2R3-MYB transcription factors [129]. The overexpression of *NtMYB3* (an R2R3-MYB from narcissus) in transgenic tobacco inhibits flavonoid biosynthesis by decreasing the expression level of the *FLS* gene [130]. In *Ginkgo biloba*, *GBMYBF2* significantly inhibits the expression of structural genes (*GBPAL*, *GBFLS*, *GBANS*, and *GBCHI*) and thus reduces the accumulation of flavonoids [131]. In soybean, *GMMYB100* negatively regulates flavonoid biosynthesis by inhibiting the promoter activities of *CHS* and *CHI* [132]. Under light conditions, the overexpression of *PPMYB17* in pear calli can activate the expression of genes associated with flavonoid biosynthesis (*PPCHS*, *PPCHI*, *PPF3H*, *PPFLS* and *PPUFGT)*, especially the expression of *FLS*; this enhances the biosynthesis of flavonoids in pear fruit [133]. FTMYB31 has been isolated from *Fagopyrum tataricum*, the overexpression of which in tobacco enhances the expression level of *CHS*, *F3H* and *FLS*, and promotes the accumulation of flavonoid biosynthesis in plants [134]. The SbMYB8 protein from *Scutellaria baicalensis* can bind to the *GmMYB92* BS3 sequence in the *SbCHS* promoter region, which positively regulates the expression of *SbCHS*, and increase the flavonoid content and antioxidant enzyme activity in transgenic tobacco [135]. The bHLH TFs play important roles in the regulation of many secondary metabolites, including flavonoids. In *citrus*, CsMYC2 is involved in the regulation of flavonoid biosynthesis through increasing the expression of *UFGT* [136]. The bHLH protein CmbHLH2 can directly bind to the promoter of CmDFR, which promotes the accumulation of flavonoids in chrysanthemum plants [137]. Interestingly, the R2R3-MYB TFs from subgroups 5, 6, and 15, bHLH from subgroup IIIf, and WD40 can interact with each other or orchestrate with others to regulate the expression of genes associated with flavonoid biosynthesis in many plants [138,139]. The TT2–TT8–TTG1 complex plays important roles in the regulation of LBGs (*DFR*, *LDOX*, *TT19*, *TT12*, *AHA10*, and *BAN*) [140]. Moreover, the MBW complex can also tissue-specifically regulate the expression of the genes involved in flavonoid biosynthesis [141]. The MYB5–TT8–TTG1 complex can regulate the expression of genes associated with flavonoid biosynthesis, such as *DFR*, *LDOX*, and *TT12*, in the endothelium, whereas the TT2–EGL3/GL3–TTG1 complex can regulate the expression of genes associated with flavonoid biosynthesis, such as *LDOX*, *BAN*, *AHA10*, and *DFR*, in the chalaza [142].

In addition, the bZIP, NAC, Dof, and WRKY TFs have also been reported to regulate flavonoid biosynthesis [143]. SlHY5, a bZIP protein from tomato, is affected by Cry1A proteins under blue light irradiation, which binds to the promoters of genes associated with flavonoid biosynthesis, such as *PAL*, *CHS1* and *CHS2*, and thereby promotes the accumulation of flavonoids [144,145]. VvibZIPC22 can activate the promoters of *VviCHS* and *VviCHI*, which increases the accumulation of flavonoids in plants [146]. The overexpression of *NtHY5* in tobacco increases the expression level of genes associated with phenylpropanoid, which can regulate the biosynthesis of flavonoids and enhance plant tolerance to salt stress [147]. The *Arabidopsis* ANAC078 increases the expression level of *CHS*, *F3’H*, *DFR*, and *LDO* genes, thereby promoting the accumulation of flavonoids under strong light conditions [148]. MdNAC52, an apple NAC TF, increases the content of flavonoids, including anthocyanins and procyanidins, in plants by binding to the promoters of *MdMYB9* and *MdMYB11*, or by binding directly to the *LAR*; both of these mechanisms further regulate the expression level of flavonoids [149]. The *Arabidopsis* AtDOF4 positively regulates the expression of late flavonoid genes (*DFR*, *LDOX* and *TT19*) and the MYB transcription factor PAP1 to promote the accumulation of flavonoids in plants [150]. In addition, the overexpression of *MdWRKY11* in apple callus promotes the expression of *F3H*, *FLS*, *DFR*, *ANS* and *UFGT*, which increases the content of flavonoids and anthocyanins [151].

### 3.3. Non-Coding RNA Regulates Flavonoid Biosynthesis

In addition, flavonoid biosynthesis is also affected by non-coding RNA, such as lncRNA (long non-coding RNA) and microRNA [152]. lncRNA may act as a precursor and endogenous target mimic of miRNA in order to indirectly regulate protein-coding genes (PCgenes) [153]. lncRNA (MSTRG.9304 and XR_001591906) can regulate the expression of the *CHS* gene in peanut seed development [154]. miPEP858a-edited and miPEP858a-overexpressing lines can regulate flavonoid biosynthesis in *Arabidopsis*, which also alters the growth and development of plants [155]. miR156 fine-tunes the anthocyanin biosynthetic pathway in poplar via the regulation of transcription factors, structural genes and other microRNAs [156].

## 4. The Roles of Flavonoid Compounds in Various Stresses

The dynamic changes in flavonoid compounds are not only regulated by plant growth and development, but are also affected by various environmental factors; these encompass both abiotic stresses, such as heavy metal stress, drought stress, UV, salt and low temperatures, and biotic stresses, including plant-parasitic nematodes, fungi and bacteria [157,158,159,160,161].

### 4.1. Biotic Stress

Various kinds of secondary metabolites that play important defense roles, including flavonoids, are produced when plants are infected with pathogens and pests and are suffering from biotic stresses [162]. Plants can be destroyed by various biotic stresses, such as bacteria, viruses, nematodes, fungi and insects [163]. Hypersensitive responses (HR) are a major element of plant disease resistance, and include the induction of lytic enzymes, the production of phytoalexins and the reinforcement of the cell wall [163,164,165]. Among these, phytoalexins are chemicals that play important roles in mediating plant responses to pests and pathogens [72]. Numerous studies have shown that some flavonoid compounds act as phytoalexins against pathogenic bacteria, fungi, and nematodes [72,166].

#### 4.1.1. The Roles of Flavonoids in the Invasion of Nematodes

Plant-parasitic nematodes can destroy the host plant by causing wounds and infecting it with microbial diseases; this destruction leads to a massive yield loss and tremendous economic losses throughout the world [167,168]. When plant-parasitic nematodes damage plants, some brown spots, galls, spots, cysts, rotting or swelling is generally formed on the roots or on the aboveground tubers [169]. Flavonoids and their derivatives, for example, coumestrol and glyceollin, play a variety of roles in the interaction between plants and nematodes, and are stimulated by nematode invasion [170,171,172]. Coumestrol, an important phytoalexin, is an isoflavonoid-like compound that can be produced in the lima bean when it is suffering from *Pratylenchus penetrans* infection [173]. Moreover, the infection of soybean roots with the root-knot nematode (*Meloidogyne incognita*) leads to an increase in the glyceollin content, which reduces the crop yield loss caused by nematodes [174]. In addition, the phytoalexin glyceollin is accumulated in the soybean root, and it has been found that there is an increase in its glyceollin content on the 2nd, 4th and 6th day after inoculation [72,175,176]. *O*-methyl-apigenin-*C*-deoxyhexoside-*O*-hexoside, a major phytoalexin in oat, has been extracted from the oat roots and shoots infected with nematodes; this was shown to reduce the invasion of cereal cyst nematodes significantly, including *P. neglectus* and *H. avenae* [72,177,178]. However, the defense mechanism of flavonoids against parasitic nematodes is still unclear.

#### 4.1.2. The Roles of Flavonoids in the Invasion of Pathogenic Fungi

Tremendous crop yield losses occur in plants when they are invaded with several kinds of plant pathogenic fungi [179,180]. To escape from various kinds of biotic stresses, plants have evolved mechanisms to resist biotic stresses, and the production of phytoalexins is an effective method for defending against the invasion of pathogenic fungi [181,182,183]. The flavonol aglycone rhamnetin is a fungi-toxic phytoalexin that enhances the resistance of cucumber to a powdery mildew [72,184,185]. There is an increase in the content of nobiletin, heptamethoxyflavone, tangeretin and sinensetin in citrus fruits when they are infected with *Phytophthora citrophthora*, which is positively correlated with their antifungal effects [186,187]. Naringenin and hesperetin have been proven to be the best antifungal compounds in flavonoids [188]. An increase in the content of isoflavonoids in tangelo Nova fruits is detected when they are treated with 6-benzylaminopurine, and the fruit resistance is enhanced by 60% when suffering from the invasion of pathogenic fungus [72,189]. In addition, the content of several kinds of phytoalexin is increased in soybean cotyledons when they are treated with four Aspergillus species, and the content of coumestrol and phytoalexins glyceollin is the highest after infection with *A. sojae* and *A. niger*, respectively [190].

#### 4.1.3. Antibacterial Effects of Flavonoids

Many researches have indicated that flavonoids can be used as antibacterial agents in plants to make them grow and develop better [191,192]. Plants can produce many kinds of phytoalexins, which are antimicrobial secondary metabolites, under various stresses, such as microbial attack [193,194]. Bean plants infected with *Pseudomonas* spp. produce antibacterial phytoalexins in bean leaves, which may enhance the plant’s resistance to *Pseudomonas* spp. [195]. It has been shown that there is an increase in coumestrol in the infected bean leaves, which blocks the bacterial colonization of *P. mars-prunorum* and *P. phaseolicola* [196]. There is also an increase in daidzein and coumestrol in soybean leaves when the plants are infected with non-pathogenic bacteria (*P. lachrymans*) and pathogenic bacteria (*P. glycinea*), indicating that coumestrol can inhibit pathogenic bacterial colonization and improve the plant’s resistance to *P. glycinea* [22,72]. Moreover, kievitone and phaseollinisoflavan also have antibacterial activity, which strongly inhibits the growth of Achromobacter and Xanthomonas species [72,197]. Recently, an isoflavonoid was proven to have a strong antagonistic effect on *Staphylococcus aureus*, which can be extracted from the roots of *Erythrina poeppigiana* [198,199].

### 4.2. Abiotic Stress

#### 4.2.1. UV Stress

Plants can rapidly produce reactive oxygen species (ROS) when they suffer under high flux, long-term UV radiation and HL (high light), which cause oxidative damage to biomolecules (DNA, proteins, RNA and membranes) [200,201]. Under UV induction and HL, plants can produce flavonoids, mainly in their epidermal cells. As different flavonoid compounds have different UV absorption capacities, flavonoids can play important roles when plants suffer from UV induction and HL, being especially useful as a protective shield [202]. Previous studies have reported that flavonols, flavones, and anthocyanins are all involved in UV and high light stress protection in plants [203]. Under UV radiation, an increase in flavonoids, such as via the production of luteolin 7-*O*-glycosides and quercetin 3-O, improves their hydroxylation levels and strengthens their antioxidant activity, which contributes to the ROS-detoxification of the plant cell and relieves the oxidative damage caused by UV stress [204,205]. Compared with the content of kaempferol under UV-B radiation, there is a higher content of quercetin and a higher ratio of quercetin/kaempferol in the leaves of some plants, which indicate that UV-B radiation could induce the substitution of the orthodihydroxy B-ring in quercetin; this would increase their adaptive dissociation [203]. Compared with low-altitude plants, high-altitude plants are subjected to high UV-B exposure. There are higher levels of rhamnosylisoorientin and maysin in the leaves of high-altitude maize lines compared to those detected in the leaves of low-altitude maize lines, which indicates that these two flavones might play important protective roles against UV-B radiation via an adaptive response [206,207,208]. Moreover, UV radiation can cause the accumulation of anthocyanin in many plants, which improves the coloration in the fruit skin of apples and other fruits [209]. Coumaroyl and mustapoyl derivatives are formed by the acylation of anthocyanins and phenylpropanoid acids, which may increase UV absorption compared to anthocyanins alone. Therefore, coumaroyl and sinapoyl derivatives might play important roles under UV-B radiation stress [203,210]. As flavonoids contain phenolic hydroxyl groups, they also exhibit antioxidant activity in the development and growth of plants [211]. The abundant flavonoids in chloroplasts are mainly dihydroxy B-ring-substituted flavonoids, such as lignan and quercetin, which play important roles in the elimination of ROS [212,213]. Under excessive light stress, the H_2_O_2_ produced by chloroplasts can be accumulated in the vesicles, which can be further scavenged by flavonoid compounds [17,214]. The ROS can be generated in different cell types and subcellular regions, which can be scavenged by many kinds of flavonols [214,215].

Many genes associated with flavonoid biosynthesis could play important roles in plants under excessive light and UV stress, including structural genes (*PAL*, *4CL*, *CHS*, *DFR*, *CHI* and *F3’5’H*) and transcription factors (MYB, bHLH, NAC, and bZIP) (Table 1). Under UV stress, the accumulation of photo-protectant flavonoids (e.g., flavones, isoflavonoids, neoflavonoids, flavanols, and anthocyanins) could absorb some harmful solar wave lengths (e.g., UV) in order to mitigate oxidative damage to cells, LDL or DNA [216,217].

#### 4.2.2. Cold Stress

Cold stresses are common stresses during the process of growth and development in plants; these include chilling stress, with temperatures of 0–20 °C, and freezing stress, with temperatures of <0 °C [244,245], both of which can seriously affect plant growth and development and reduce crop productivity, especially for crops that are sensitive to low temperatures [246]. Chilling stress can affect plant growth from germination to maturity, and can also determine the distribution of plants around the world [247]. There are significant differences among various plants regarding when and how they suffer from chilling stress. Some plants can grow well at chilling temperatures, such as some overwintering cereals and *Arabidopsis*, while other plants struggle to survive at chilling temperatures, such as many tropical and subtropical plants (maize, rice and tobacco) [248,249]. Previous studies have reported that plants undergo a series of physiological and cellular changes under low-temperature conditions, including changes in calcium signal, photosynthesis, metabolism and membrane structure [248,250]. Freezing stress is much more harmful to plants than low-temperature stress, and may even lead to plant death. Under natural conditions, freezing damage starts from extracellular ice nucleation [251,252]. Once ice nuclei are formed, they gradually become larger and form ice crystals, which diffuse into the apoplast. In the apoplast, ice crystals can induce water outflow, which can lead to cell dehydration. When ice crystals diffuse into cells, irreversible damage occurs [252,253]. Flavonoids play an important role in coping with freezing stress. Schulz et al. [254] revealed that flavonoids are the determining factor in the freezing resistance and cold adaptation of *Arabidopsis*, which suggests that the complete loss or substantial reduction in flavonoids leads to the damage of the freezing resistance mechanism. Previous studies have reported that several flavonoids (flavones, flavonols, flavanols, and anthocyanins) display a strong resistance to cold stress. In *Cryptomeria japonica*, cold stress can enhance the production of anthocyanins, flavonoids and flavonols through the up-regulation of *FLS*, which provides a good basis for the molecular mechanism of response to cold stress in spruce [229]. In addition, an increase in anthocyanins in their epidermal cells can decrease the osmotic potential of cells and delay freezing through the surface nucleate [255]. When *Fagopyrum tataricum* was subjected to cold stress, the total content of anthocyanins in the epidermis and cortex cells of hypocotyl was twice those in cotyledons, which improved their cold resistance [233]. Moreover, many genes associated with flavonoid biosynthesis are up-regulated when suffering from cold stress, including structure genes (*CHI*, *CHS*, *C4H*, *ANS*, *UFGT*, *F3H*, and *DFR*) and TFs (MADS-box, bZIP, MYB, and bHLH), further confirming that flavonoids play a key role in enhancing cold resistance [233,234,235] (Table 1).

#### 4.2.3. Salt Stress

Salinity stress is a major threat to global food production, which can inhibit plant growth and development through osmotic stress, cytotoxicity caused by the excessive absorption of Na^+^ and Cl^−^, and nutritional deficiencies [256,257,258]. Research shows that 20% of the world’s irrigated land suffers from excess soluble salts [259]. The limiting effect of excess soluble salts in the soil on plant growth and development is mainly mediated through two mechanisms: osmotic stress and ion toxicity [260]. The increased ion concentration in soil causes a low solute/osmotic potential, reduces the ability of plant roots to uptake water, and eventually blocks plant growth and development [72]. Moreover, ion toxicity caused by sodium accumulation damages the cell membrane and disturbs various plant physiological processes, such as photosynthesis, respiration, transpiration and osmoregulation, which eventually cause plant necrosis or chlorosis [72,261]. Yan et al. [224] found that NaCl treatment significantly increases the content of flavonoid glycosides through the up-regulation of *GMFNSII-1* and *GMFNSII-2* in *Glycine* max, indicating that flavonoid glycosides play a positive role in enhancing salt tolerance. Salt stress also increases the luteolin content in *Cajanus cajan* by upregulating the expression level of *CcPCL1* and *CcF3′H-5*, which greatly enhances the salt tolerance of plants [262]. Moreover, the content of flavonoids accumulates differentially between the roots and leaves of *Solenostemma argel* under salt stress; this improves the antioxidant capacity and osmoregulatory capacity of plants under salt stress [263]. Wang et al. [264] found that high salt stress (85 mmol/L) promotes anthocyanin biosynthesis in cultivated purple sweet potato, and that low salinity has a significant effect on the biosynthesis of phenolic acids and flavonols in the plant. Under low salt stress, plants can regulate genes to control the concentrations of cellular Na^+^ and K^+^ in plasma; however, they lose this ability under high salt stress. In addition, it has also been found that salt stress increases the content of isoflavones by regulating the expression of *GMIFS1*, which improves the antioxidant capacity and osmotic stress tolerance of soybean [228].

#### 4.2.4. Drought Stress

Drought is a major environmental stress factor that affects various physiological and biochemical processes in plants [265,266]. To escape from environmental stresses, plants have evolved complex mechanisms to respond to stress at the physiological and biochemical levels, including the production of compatible osmolytes, endogenous hormonal changes, and molecular changes [267,268]. The plants mediate the balance between growth and resistance to stress by modulating the architecture of the root system and stomatal closure under moderate drought stress [269,270]. Plants can induce cellular protection against severe drought stress, modulate the antioxidant enzyme systems to remove ROS, and accumulate proteins to maintain cell turgor [271].

Flavonoids, including anthocyanins, flavonols, flavanols, and flavones, play a major role in the response to drought stress. Under drought stress conditions, plants can accumulate anthocyanins to protect them against excessive sunlight and prevent water loss by reducing stomatal transpiration and density; this allows them to survive in a severe drought environment [272]. The biosynthesis of catechin in the tea plant was shown to be induced under drought stress, increasing the plant’s ROS scavenging capacity and enhancing its drought stress tolerance [273]. There was an increase in cyanidin, delphinidin 3-*O*-glucoside, and cyanidin 3-*O*-glucoside in the calyx of Roselle Cultivars at physiological maturity under 35% water-loss stress [274]. There was an increase in the flavonoid content, total phenolic capacity and antioxidant capacity under water deficit stress in Peppermint [275]. Moreover, Mechri et al. [276] found that the concentrations of catechin, quercetin, luteolin 7-*O*-glucoside, and apigenin 7-*O*-glucoside in olive leaves increased after drought treatments, indicating that these phenolic compounds regulate the olive water status and reduce the oxidative damage caused by water deficit stress.

Numerous studies have shown that genes associated with flavonoid biosynthesis are regulated independently by drought stress, such as *PAL*, *CHS*, *CHI*, *F3H*, *DFR*, *ANS*, *NAC*, WRKY, bZIP, MYB, and bHLH (Table 1). For example, the expression level of *TaCHS*, *TaCHI*, *TaF3H*, *TaFNS*, *TaFLS*, *TaDFR* and *TaANS* in wheat has been shown to rapidly increase after drought stress, thus increasing the total contents of phenols, flavonoids, and anthocyanins [221,222,223]. Transcriptome analysis has indicated that several TFs (bHLH, NAC, MYB, and WRKY) play important roles in response to cold and drought stress during seed germination [242]. In addition, the transcriptome analysis of *Cicer arietinum* treated with drought stress was conducted, and the results showed that AP2-EREBP, bHLH, bZIP, MYB, NAC, WRKY and MADS were involved in the drought stress response [241].

#### 4.2.5. Heavy Metal Stress

In recent decades, the rapid development of industry and the use of pesticides have caused heavy metal contamination in the soil [277]. Heavy metal stress has notable adverse effects on crop productivity and growth [278,279]. Heavy metals, including lead (Pb), cadmium (Cd), nickel (Ni), cobalt (Co), zinc (Zn), chromium (Cr), iron (Fe), arsenic (As), and silver (Ag), are present in dispersed form in rock formations, and are considered to be one of the potential threats to crop plant productivity [280,281]. Although all heavy metals are non-biodegradable and immobile, heavy metals can be absorbed by plant root systems via diffusion, endocytosis or through metal transporters [282]. Heavy metal stress causes the inactivation of enzymes and interferes with the substitution reactions of essential metal ions from biomolecules [279,283]. Heavy metal stress destroys the membrane integrity and alters the basic plant metabolic reactions, such as photosynthesis, respiration and homeostasis [284]. Moreover, heavy metal stress generates ROS, such as superoxide radical (O^−2^), hydroxyl radical (OH) and hydrogen peroxide (H_2_O_2_), and the cytotoxic compound, which leads to lipid peroxidation; this damages biomolecules and destroys DNA strands [285].

In order to escape from environments containing heavy metals, plants have gradually evolved mechanisms to cope with these heavy metals. The functional roles of flavonoids in the response to heavy metal stress have received some attention [286]. For example, studies have shown that the contents of flavonols (quercetin and kaempferol) increases in *Arabidopsis* via the upregulation of *FLS1* under lead stress (Pb) treatment, thus alleviating lead toxicity [230]. Cd stress has been shown to induce the accumulation of isoflavone compounds such as malonylolonin, medicarpin, coumestrol, formonetin, and medicarpin3-*O*-b-(60-malonylglucoside) in *Medicago truncatula* roots after Cd stress, thus alleviating Cd toxicity [287]. Moreover, Cd stress has been shown to increase the accumulation of anthocyanins via the upregulation of *CHS* and *DFR* in *Azolla imbricata*, thus improving their ROS scavenging ability and enhancing the tolerance of *Azolla imbricata* to Cd stress [288].

## 5. Transgenic Technologies Used in Enhancing Stress Resistance

In agricultural biotechnology, many techniques are used to alter the genetic structure of plants to produce genetically modified plants [289]. Transgenic technology can be used to improve plant traits (yield and quality), as well as to solve agricultural problems (biotic and abiotic stresses) [290,291].

### 5.1. Transgenic Technology Used in Enhancing UV and HL Resistance

Transgenic plants that carry structure genes and TFs for UV/HL tolerance are being developed, mainly by using *Agrobacterium* methods [292]. The overexpression of the *CHS* gene has been shown to enhances resistance to high levels of light in *Arabidopsis* leaves by increasing the content of anthocyanins [219]. The overexpression of the *MnFNSI* gene from *Morus notabilis* in tobacco has been shown to increase the content of flavones in leaves and enhance their tolerance to UV-B stress [293]. The heterologous expression of *Pn2-ODD1* in *Arabidopsis* has also awarded plants tolerance to UV-B radiation and oxidative stress by increasing their antioxidant capacity [294]. *Arabidopsis* lacking *ANS* have been shown to be sensitive to high light levels due to the impairment of anthocyanin photoprotection [295]. Moreover, TFs involved in the flavonoid biosynthesis have been found to play important roles under UV stress. The overexpression of *GmMYB12B2* in *Arabidopsis* promotes flavonoid biosynthesis and improves the tolerance of transgenic *Arabidopsis* to salt and UV stress [236]. In transgenic apple, MdWRKY72 and MdBBX20 enhance the plant resistance to UV-B stress through the regulation of genes associated with anthocyanin biosynthesis (*MdANS*, *MdDFR*, *MdUFGT*, and *MdMYB1*) [243]. The overexpression of *ANAC078* in *Arabidopsis* increases the content of anthocyanin, and enhances the tolerance of plants to HL stress, which is opposite to the *ANAC078* mutant plants [148].

### 5.2. Transgenic Technology Used in Enhancing Salt Resistance

Transgenic plants that carry structure genes and TFs for salt tolerance are currently being developed. For example, the overexpression of *EaCHS1* in tobacco has been shown to increase the production of downstream flavonoids and the expression of related genes in the phenylpropanoid pathway, and to maintain ROS homeostasis; this improves the plant’s resistance to salinity stress during seed germination and root development [220]. The overexpression of *AtDFR* in Brassica napus increases the content of anthocyanins, which enhances its salt tolerance in comparison with wild-type plants under high-salt stress conditions [231]. Moreover, transgenic *Arabidopsis* overexpressing the wheat *TaDFR-I* gene augments the accumulation of anthocyanin and promotes the tolerance of plants to high salt penetration [232]. Transgenic *Arabidopsis* plants overexpressing PnF3H significantly increase the expression level of resistance genes (*AtSOS3*, *AtP5CS1*, *AtHKT1*, *AtCAT1* and *AtAPX1*), reduce the hydrogen peroxide content and enhance the salt and oxidative stress tolerance of plants [225]. Moreover, many TFs associated with flavonoid biosynthesis, such as WD40, MYB, bHLH, bZIP, NAC, WRKY, and MADS-box, played important roles in response to salt stress [223,296]. Transgenic *Arabidopsis* overexpressing TaWD40 from *Triticum aestivum* significantly improves the tolerance of plants to salt osmotic stresses [240]. The overexpression of grapevine *VvbHLH1* in *Arabidopsis* increases the flavonoid contents and enhances the salt tolerance of plants [237].

### 5.3. Transgenic Technology Used in Enhancing Drought Resistance

Transgenic plants carrying structure genes and TFs for drought tolerance have been reported [297]. For example, transgenic tobacco overexpressing the *IbC4H* gene from the purple sweet potato has been shown to significantly promote the biosynthesis of flavonoids such as anthocyanins and flavonols, enhance the ROS scavenging capacity of plants, and improve their drought resistance [218]. The overexpression of *Lycium chinense*
*LcF3H* in tobacco also increases the flavan-3-ol content (including catechin and epicatechin) and improve the antioxidant system, which strengthens the drought stress tolerance of the plant [226]. In addition, the overexpression of *EcbHLH57* from *Eleusine coracana* in tobacco leads to the accumulation of more flavonoids, such as flavonols and flavanols, and enhances the photosynthetic rate and stomatal conductance; these factors significantly improve the tolerance of the plant to salt stress and drought stress [238]. The transgenic *Arabidopsis* that overexpresses *ZmWRKY40* promotes the biosynthesis of flavonoids, increases POD and CAT activities, and reduces the ROS content, which improves the drought stress tolerance of the plant [298].

### 5.4. Transgenic Technology Used in Enhancing Low Temperature Resistance

Low temperatures significantly increase the content of flavonoids via the regulation of genes involved in flavonoid biosynthesis [299]. The overexpression of *SlF3H* in tobacco plants strongly induces the expression of *CHS*, *CHI*, and *FLS*, and increases the contents of anthocyanins and other flavonoids; this improves the tolerance of the plant to cold stress [227]. Transgenic *Arabidopsis* that overexpress *OrbHLH001* has been shown to increases the accumulation of flavonoids such as flavonols, anthocyanins, and flavanones through the regulation of *CHS*, *CHI*, *F3H*, and *C4H*, which enhance the tolerance of transgenic *Arabidopsis* to freezing and salt stress [239]. The overexpression of *SlNAM1* in tobacco plants accumulates many more anthocyanins and other flavonoids, thus improving the cold tolerance of the plants [300].

## 6. Conclusions and Future Prospects

The biosynthesis of flavonoids is the most studied pathway of secondary metabolism in plants. The defensive role of flavonoids in plants against various biotic and abiotic stresses was reviewed in Figure 3, and their significant contributions to plant resistance were also discussed. Flavonoids contain several subgroups, including anthocyanidins, flavonols, flavones, flavanols, flavanones, chalcones, dihydrochalcones and dihydroflavonols, which are widely distributed in various plants. However, the detailed functions of particular flavonoids are still not clear, and need to be explored. The roles of specialized flavonoids in response to various stresses are also worth evaluating. With the rapid advancements made in sequencing technology, there have been major achievements regarding the identification of candidate genes associated with flavonoid biosynthesis. Although many genes have been identified, some specific genes associated with certain important flavonoids are poorly understood, and need to be studied further. Although many researchers have explored the functions of genes associated with flavonoid biosynthesis through transgenic approaches, it may take a long time to verify their gene functions. It is evident that flavonoids play important protective roles against harmful biotic and abiotic stresses, but that the knowledge of flavonoids under various stresses is still incomplete and requires extensive research.

## Figures and Tables

**Figure 1 molecules-28-03599-f001:**
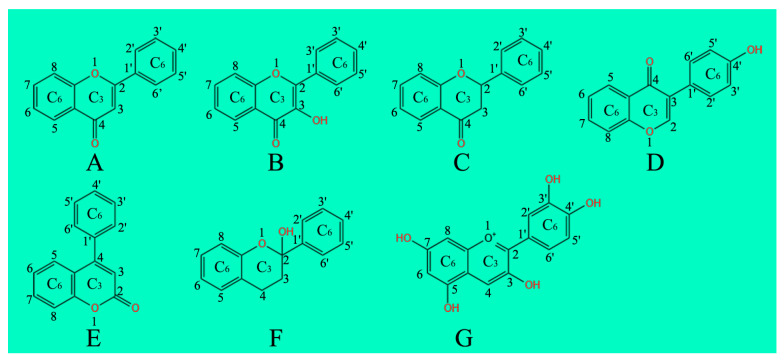
Chemical structures of flavonoid compounds. (**A**) Flavones; (**B**) Flavonols; (**C**) Flavanones; (**D**) Isoflavonoids; (**E**) Neoflavonoids; (**F**) Flavanols; (**G**) Anthocyanins.

**Figure 2 molecules-28-03599-f002:**
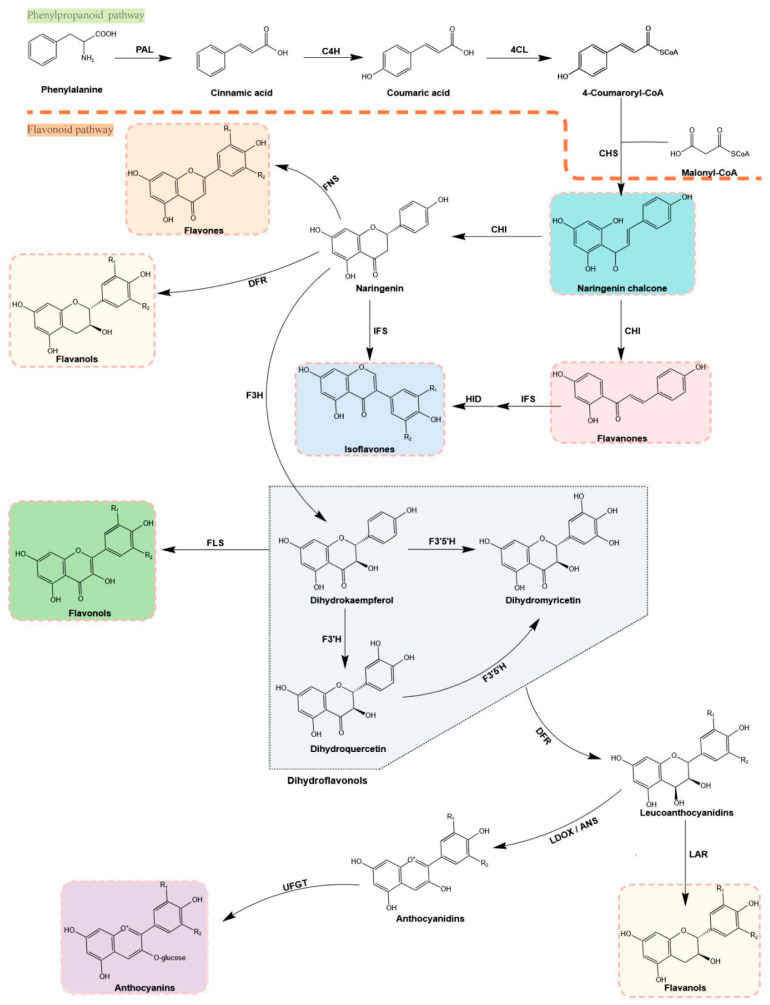
The pathway of flavonoid biosynthesis.

**Figure 3 molecules-28-03599-f003:**
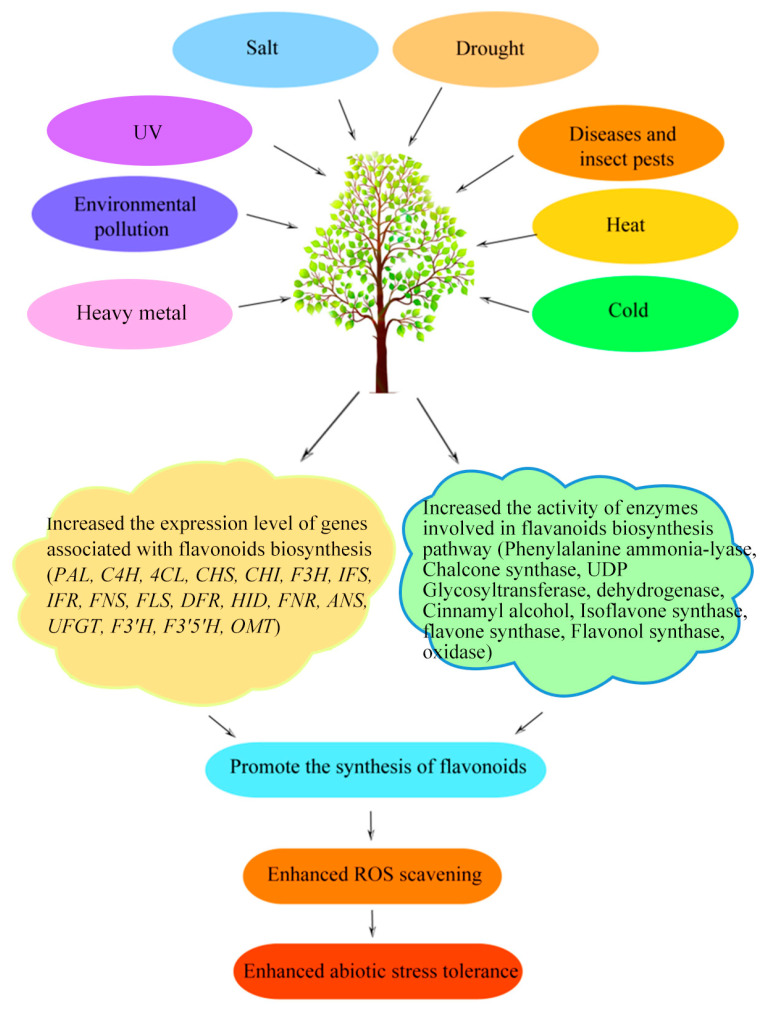
Molecular mechanisms of flavonoids in response to biotic and abiotic stresses.

**Table 1 molecules-28-03599-t001:** The roles of genes involved in flavonoid biosynthesis and the response of plants to various biotic and abiotic stresses.

Gene Name	Expression Status	Roles in Biosynthesis	Stress Resistance in Plants	References
Structural genes				
*PAL*	High expression under stress	Involved in the biosynthesis of flavonoids	Played positive roles in response to abiotic stress	[16,72,104,124]
*C4H*	High expression under stress	Involved in the biosynthesis of flavonoids	Played positive roles in response to abiotic stress	[105,216,217,218]
*4CL*	High expression under stress	Involved in the biosynthesis of flavonoids	Played positive roles in response to abiotic stress	[106,216,217]
*CHS*	High expression under stress	Involved in the biosynthesis of flavonoids	Played positive roles in response to abiotic stress	[108,219,220]
*CHI*	High expression under stress	Involved in the biosynthesis of flavonoids	Played positive roles in response to abiotic stress	[109,221,222,223]
*FNS*	High expression under stress	Involved in the biosynthesis of flavones	Played positive roles in response to UV and salt stress	[224]
*F3H*	Constitutive of high expression in transgenic *Arabidopsis* and *tobacco*	Involved in the biosynthesis of flavan-3-ol (catechin and epicatechin)	Played positive roles in response to salt, drought and cold stress	[225,226,227]
*F3’H*	High expression under stress	Involved in the biosynthesis of flavonoids	Played positive roles in response to abiotic stress	[112,113,114,120,121]
*F3’5’H*	High expression under stress	Involved in the biosynthesis of flavonoids	Played positive roles in response to abiotic stress	[112,113,114,115,216,217]
*IFS*	High expression under stress	Involved in the biosynthesis of isoflavones	Played positive roles in response to salt osmotic stress	[228]
*FLS*	High expression under stress	Involved in the biosynthesis of flavonols (kaempferol, quercetin, and myricetin)	Played positive roles in response to UV stress, salinity stress, drought stress, cold stress and heavy metal stress	[114,133,215,229,230]
*DFR*	Constitutive of high expression in transgenic *Arabidopsis*	Involved in the biosynthesis of leucoanthocyanidin and anthocyanins	Played positive roles in response to salt, cold and UV stress	[114,231,232]
*ANS*	High expression under stress	Involved in the biosynthesis of anthocyanidin	Played positive roles in response to abiotic stress	[151,203,233,234,235]
*UFGT*	High expression under stress	Involved in synthesis of stable anthocyanins	Played positive roles in response to abiotic stress	[120,121,151,233,234,235]
Transcription factors				
*MYB*	Constitutive of high expression in transgenic *Arabidopsis* and *Petunia*	Regulated of anthocyanins and other flavonoids biosynthesis	Played positive roles in response to UV, salt and heavy metal stress	[236]
*bHLH*	Constitutive of high expression in transgenic *Arabidopsis* and tobacco	Regulated of flavonols, flavanols, anthocyanins, and flavanones biosynthesis	Played positive roles in response to drought, freezing and salt stress	[237,238,239]
*WD40*	Constitutive of high expression in transgenic *Arabidopsis*	Regulated of flavonoids biosynthesis	Played positive roles in response to ABA and salt osmotic stress	[240]
*bZIP*	Constitutive of high expression in transgenic *tobacco*	Regulated of flavonoids biosynthesis	Played positive roles in response to High light, UV and salt stress	[147,216,217]
*NAC*	high expression; Constitutive of high expression in transgenic *Arabidopsis*	Regulated of anthocyanins and procyanidins biosynthesis	Played positive roles in response to High light and UV stress	[143,148,149]
*MADS-box*	High expression under stress	Regulated of flavonoids biosynthesis	Played positive roles in response to cold, salt, and drought stress	[223,233,234,235,241]
*WRKY*	High expression under stress; Constitutive of high expression in transgenic apple	Regulated of anthocyanins and other flavonoids biosynthesis	Played positive roles in response to cold, drought, and UV stress	[151,242,243]

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
