# Peer review of "The Classification, Molecular Structure and Biological Biosynthesis of Flavonoids, and Their Roles in Biotic and Abiotic Stresses"

_molecules, 2023, doi:10.3390/molecules28083599_

Round 1
Reviewer 1 Report
The review by Zhuang and co-workers is a competent paper, with a useful content. I have only minor suggestions.
1. Title does not fully reflects the content. Flavonoids in stress started only from the chapter 4. Please, consider a more concise title.
2. Chapter 4.1.1. It is not clear, whether flavonoids play role in the defence or their production is just a collateral factor. Please, clarify/prove.
3. Chapter 4.2.3. There are many redundancies (repeating of the same in other words) at the beginning of the chapter. Pleease, take care.
4. There are many language issues (see highlighted .pdf in the attachment). Perhaps, I missed some- please, do not limit to my corrections only. The misuse of tenses is the most common. Please, consider that the Past tense reflects something, which happened once in the past (and, perhaps, not anymore). So, if this is used in the context of "found", "proved", "reported", "showed" etc it is OK. But if you refer to some conclusion resulted from a study, consider the usage of the Present tense. Examples: "Heavy metal stress caused the inactivation of enzymes and interfered with...", "Flavonoids contained several subgroups including..."- should be set in the Present tense. Alternatively, if you doubt a particular report and do not want to absorb the responsibility for its credibility, you should quote it as "it has been reported that.." or "X and co-workers found that..."

Reviewer 2 Report
The review "Molecular mechanisms of flavonoids on the response to various biotic and abiotic stresses in plants" by Zhuang et al. gives a general vision of the possible roles of flavonoids against stress. As most of the molecular mechanisms for flavonoid function are unknown (as it is reviewed in this paper), the title can be changed to "Involvement of flavonoids..."or "The role of flavonoids..."
Reading of section 2 (Flavonoids classification) should be helped if carbon numbering and ring nomenclature would be included in some of the structures shown in Figure 1.
Page 3, flavanones: a double bond is not a saturated bond, therefore sounds strange the expression "the saturated double-bond".
Page 11: There are more appropiate references than 212,213 to support the sentence "The most abundant flavonoids in chloroplasts..."See, for instance, Agati et al (2012) Plant Sci196: 67-76 and references therein. There are some flaws in the same page: "in the cell of vesicles" has not sense; neither "in different cell types such as chloroplasts"; nor "could reduce some harmful solar wavelenghts" (in this last sentence is better to use the term "absorb")
English and typing must be reviewed in the whole ms. For instance (page 15), the sentence "Transgenic Brassica napus overexpressing ArabidopsisAtDFR were able to synthesize and accumulate more anthocyanins, which have higher chlorophyll content and salt tolerance in comparison with wild-type plants under high salt medium stress conditions" is very confusing. This kind of sentences are spreaded all along the text.
Page 16: The roles of flavonoids could be schematized in Fig 3, but, by sure, are not reviewed or discussed in this figure.
The meaning of table 1 is not clear.What do you mean by high expression? Constitutive high expression or high expression under stress? It is not well-understandable the expression "control the synthesis of flavonoids" when you refer to the genes that codify the enzymes implied in the biosynthesis of these compounds. What do you want to mean in the last column with "resistance to" Are you referring to overexpressed genes?
Reviewer 3 Report
The manuscript is a review that systematically describes the molecular function of flavonoids in the response to biotic and abiotic stresses. In my opinion, the review is very well thought out, and well summarize the results of the analysis of literature data described in the text. This article made a positive impression on me.
Minor comments:
Keywords should be in alphabetic order General remark – short name of genes should be written in italic, short name of protein should be written using capital letters, and no italic – please change in the manuscript for example page 12 and others In manuscript sometimes authors used Present Simple and sometimes Past Simple to describe the results of other. In my opinion it is better to use past tense. Page 6 – in the title of section 3.2 - please add abbreviation: TFs Page 8, line 5 - there is increased – should be increased Page 11, line 5 counting from the bottom should Schulz et al[228]… Page 13, at the subsection 4.2.4 – in the first paragraph - I don’t know if information refers to water deficit stress or stress in general Page 13: there is …in the calyx of Roselle Cultivars - is it OK? Figure 3: Please leave only these stresses about which you wrote i.e waterlogging could be delete; Information in the clouds is hardly legibleAuthor Response
Please see the attachment.
